# Implications of Flagellar Attachment Zone Proteins TcGP72 and TcFLA-1BP in Morphology, Proliferation, and Intracellular Dynamics in *Trypanosoma cruzi*

**DOI:** 10.3390/pathogens12111367

**Published:** 2023-11-18

**Authors:** Normanda Souza-Melo, Carolina de Lima Alcantara, Juliana Cunha Vidal, Gustavo Miranda Rocha, Wanderley de Souza

**Affiliations:** 1Laboratório de Ultraestrutura Celular Hertha Meyer, Centro de Pesquisas em Medicina de Precisão, Instituto de Biofísica Carlos Chagas Filho, Universidade Federal do Rio de Janeiro, Rio de Janeiro 21491-590, Brazil; alcantara@biof.ufrj.br (C.d.L.A.); julianavidal@biof.ufrj.br (J.C.V.); mirandarocha@gmail.com (G.M.R.); 2Centro de Estudos Biomédicos-CMABio, Escola Superior de Saúde, Universidade do Estado do Amazonas-UEA, Manaus 69065-000, Brazil

**Keywords:** flagellar attachment zone, morphological changes, CRISPR/Cas9

## Abstract

The highly adaptable parasite *Trypanosoma cruzi* undergoes complex developmental stages to exploit host organisms effectively. Each stage involves the expression of specific proteins and precise intracellular structural organization. These morphological changes depend on key structures that control intracellular components’ growth and redistribution. In trypanosomatids, the flagellar attachment zone (FAZ) connects the flagellum to the cell body and plays a pivotal role in cell expansion and structural rearrangement. While FAZ proteins are well-studied in other trypanosomatids, there is limited knowledge about specific components, organization, and function in *T. cruzi*. This study employed the CRISPR/Cas9 system to label endogenous genes and conduct deletions to characterize FAZ-specific proteins during epimastigote cell division and metacyclogenesis. In *T. cruzi*, these proteins exhibited distinct organization compared to their counterparts in *T. brucei*. TcGP72 is anchored to the flagellar membrane, while TcFLA-1BP is anchored to the membrane lining the cell body. We identified unique features in the organization and function of the FAZ in *T. cruzi* compared to other trypanosomatids. Deleting these proteins had varying effects on intracellular structures, cytokinesis, and metacyclogenesis. This study reveals specific variations that directly impact the success of cell division and differentiation of this parasite.

## 1. Introduction

The pathogenic protozoan *Trypanosoma cruzi* causes Chagas disease and has developed efficient methods for exploiting host organisms (both vertebrates and invertebrates), which involve various developmental stages. As a result, *T. cruzi* undergoes significant morphological modifications, including alterations in gene expression and rearrangements of its structures and organelles [1,2,3,4,5,6]. The morphological identification of the epimastigote, trypomastigote, transitional, and amastigote forms of *T. cruzi* is mainly based on the relative positioning of the nucleus, kinetoplast, flagellum, and the length of the flagellum due to the 3D arrangement of the subpellicular microtubules [3,6,7,8] in conjunction with its molecular and biochemical composition [9,10,11].

*T. cruzi* has a single flagellum that is responsible for parasite motility and is composed of a classic conserved axoneme (9 + 2 arrangement). However, like most of the members of the Trypanosomatidae family, it has a unique filamentous structure known as the paraflagellar rod (PFR), which is closely associated with the axoneme [12,13,14,15]. In most trypanosomatids, including *Trypanosoma brucei* and *Leishmania* spp., the flagellum departs from the flagellar pocket, an extracellular space formed by the invagination of the plasma membrane. It is responsible for endocytic and exocytic activities [16,17]. Distinctively, *T. cruzi* does not exhibit significant endocytosis via the flagellar pocket; instead, it has developed a highly specialized cytostome–cytopharynx complex that is responsible for this process, constituting the most polarized endocytic system described thus far [18,19,20,21].

After it exits the flagellar pocket, the flagellum remains attached to the cell body until it becomes free at the cell anterior. At this location, a special type of junction is formed between the flagellar membrane and the cell body membrane. One special feature in trypanosomatids that belong to the Trypanosoma genus is a large area where there is a close association of the flagellum with the cell body. This structure was initially characterized using conventional transmission electron microscopy (TEM) of thin sections [22] and freeze-fracture replicas [19,20,21]. This highly specialized area was designated as the flagellar attachment zone (FAZ), which also presents a set of cytoplasmic fibers, filaments, and junctional complexes that start from the flagellar domain and are anchored to the cell cytoskeleton [22,23,24,25,26]. 

While well-characterized in *T. brucei*, the FAZ remains poorly understood in *T. cruzi*. It is a key regulator of cellular processes such as cell motility, division, morphogenesis, and infectivity [17,24,27,28]. It comprises an electron-dense cytoplasmic region composed of the FAZ fibrillary network. This network involves several domains: the FAZ flagellar domain, FAZ intercellular domain, FAZ filament domain, microtubule quartet domain, and microtubule quartet–FAZ linker domain [22,24,28]. This complex also includes the special organization of the flagellar and cell body plasma membrane, which was characterized using freeze-fracture, and a special organization of intramembranous particles was identified [19,20,21].

In *T. brucei*, 34 proteins were identified as part of the FAZ, and the majority have at least one orthologue in *Leishmania mexicana* [17,25]. These proteins are heterogeneously distributed along the FAZ length and flagellum. It has been shown that protein function depends on localization, and alterations in components in the same domain can lead to similar phenotypes. In trypanosomes, the first molecule characterized in *T. cruzi* was glycoprotein 72 (GP72), a membrane glycoprotein of 72 kDa [29]. It was localized along the flagellum using an immunofluorescence assay, and TcGP72-null mutants showed detachment of the flagellum from the cell body, changing cell morphology, and reducing the ability to infect insect vectors and mammalian cells [15,29,30]. 

Despite the well-established depletion phenotype of TcGP72, little is known about its interaction with other proteins and their structural features during the life cycle of *T. cruzi*. TcGP72 protein is a homolog of flagellum adhesion protein 1 (FLA-1), FLA-2, and FLA-3 in *T. brucei* [31]. These proteins are structured with an N-terminus peptide signal, an extracellular region with several N-glycosylation sites, an NHL-repeat domain, a transmembrane domain, and a short intracellular C-terminus [27,32]. In *T. brucei*, TbFLA1 protein interacts with FLA1-binding protein (TbFLA-1BP), promoting membrane adhesion between the flagellum and the cell body. TbFLA1 is localized on the side of the cell body and anchored to the membrane by the C-terminus transmembrane domain, which is a short tail comprising 16 amino acids. It is responsible for correctly targeting the extracellular domain and accomplishing the interaction of TbFLA1 with TbFLA-1BP. This TbFLA-1/TbFLA-1BP association favors the assembly of the FAZ and flagellum and regulates cell morphogenesis [27].

In this study, we examined the role of the FAZ proteins TcGP72 and TcFLA-1BP using tagged parasites and null mutants for both proteins using a CRISPR/Cas9 system in epimastigote forms (the replicative stage in the insect vector) and during metacyclogenesis. We report that the arrangement of FAZ proteins in *T. cruzi* differs from their counterparts in *T. brucei*, with fewer isoforms compared to other Trypanosoma species. We demonstrate that the TcFLA-1BP protein is distributed along the parasite’s cell body, exhibiting a reverse localization compared to what has been previously described in *T. brucei*.

The TcGP72 mutant labeled with mNeonGreen at the C-terminus acted as a dominant-negative mutant, displaying flagellar detachment similar to the null mutant. Consistent with the previous studies on *T. brucei* and *T. cruzi*, we observed varying degrees of flagellar detachment from the cell body in the null mutants for the evaluated proteins. We show that the complete or partial disruption of the FAZ compromised cell morphometry, resulting in cells with an altered cell body and flagellum lengths. 

The absence of TcGP72 and TcFLA-1BP altered cell growth and differentiation, significantly impacting cytokinesis, with a substantial number of parasites in the G2/M phase displaying aberrant forms. Our data indicate that despite the phenotypes found in the mutant parasites being similar to those described in *T. brucei*, *T. cruzi* exhibits unique characteristics in FAZ structure formation that may contribute to the success of cell division and parasite differentiation. Understanding the arrangement of these components in the parasite may contribute to better comprehension of changes that take place during the life cycle of this protozoan.

## 2. Materials and Methods

### 2.1. Parasite and Mammalian Cell Maintenance

*T. cruzi* epimastigotes (Dm28c strain) were maintained in the exponential growth phase in liver infusion tryptose (LIT) medium [33] supplemented with 10% fetal bovine serum (FBS), 10 U/mL of penicillin, and 10 μg/mL of streptomycin (Invitrogen, Carlsbad, CA, EUA) at 28 °C. The parasites were usually seeded at 5 × 10^6^ cells/mL (logarithmic growth phase) and counted manually in a Neubauer chamber. For metacyclogenesis, epimastigotes were grown until they reached a stationary phase (5 × 10^7^ cells/mL), and then differentiation was accomplished as described previously [34]. 

Metacyclic trypomastigotes (MTs) were purified from the supernatant with ion-exchange chromatography using 2-(diethylamino) ethyl ether cellulose (DEAE-cellulose) columns [35]. Tissue-culture cell-derived trypomastigotes (TCTs) were obtained from supernatants of LLC-MK2 cells (Rhesus monkey kidney epithelial cells, ATCC CCL-7). The cells were infected with MTs or TCTs released from infected cells. The parasites were collected with centrifugation at 1500 g for 5 min and incubated for at least 60 min at 37 °C. TCT-enriched supernatants were collected and used for experiments.

### 2.2. T7Cas9 Line Generation

*T. cruzi* Dm28c T7Cas9 line was generated with electroporation using an Amaxa Nucleofector 2b (Lonza). This was performed by following the manufacturer’s recommendations with two pulses of X-014 program in electroporation buffer (5.0 mM KCl, 0.15 mM CaCl_2_, 90 mM Na_2_HPO_4_, 50 mM HEPES, and 50 mM Mannitol) [36]. For expression of Cas9 and T7 RNA polymerase (RNAP), the pLEWT7Cas9-Neo plasmid was linearized with Not I before transfection. Further, 2 × 10^7^ cells of wild-type *T. cruzi* Dm28c epimastigotes were transfected with 50 μg of linearized pLEWT7Cas9-Neo plasmid. After transfections, the parasites were positively selected [37].

### 2.3. Endogenous C-Terminus Tagging with CRISPR/Cas9 Editing

For C-terminus tagging of endogenous proteins, we used the Dm28c T7Cas9 line and a specific single guide RNA (sgRNA) sequence targeting the 3′ end of two different genes using the genome sequence of *T. cruzi* Dm28c 2018 (accessed 12 March 2020. These genes were TcGP72 (gene ID C4B63_21g104; flagellum adhesion protein 1 or GP72), a homolog to FLA1, FLA2, and FLA3 in *T. brucei* (gene ID Tb927.8.4010, Tb927.8.4060, and Tb927.8.4110, respectively), and TcFLA-1BP (gene ID C4B63_21g106, FLA1-binding protein), an orthologue to a gene in *T. brucei* (gene ID Tb927.8.4100). Each of these two mutants was generated with co-transfection of one SgRNA template with a DNA donor cassette to induce homology-directed repair in epimastigotes of the T7Cas9 line. The design of the primers used for the amplification of the sgRNA template and donor DNA was conducted through the utilization of the LeishEdit platform http://www.leishgedit.net/Home.html (accessed on 25 June 2020), using the genome of the *T. cruzi* strain DM28c.

To determine the sgRNA targeting sequence, the Eukaryotic Pathogen CRISPR Guide RNA/DNA Design ToolEuPaGDT http://grna.ctegd.uga.edu (accessed on 25 June 2020) was used to select the best protospacer sequence. Moreover, 3′ SgRNA targeting was designed to induce a double-stranded break with Cas9 nuclease downstream of the stop codons of each gene. Chimeric 3′ sgRNA templates were obtained with PCR with a forward primer (3′ SgRNA F) containing the T7 RNAP promoter (taatacgactcactatagg) for in vitro transcription of RNA, target sequence (PROTOSPACER), and a region complementary to a universal reverse primer G00 containing the guide RNA scaffold sequence for Cas9 from Streptococcus pyogenes (SpCas9). sgRNA primer designer: T7 RNAP promoter/PROTOSPACER/region complementary RNA scaffold SpCas9. G00 universal SgRNA (aaaagcaccgactcggtgccactttttcaagttgataacggactagccttattttaacttgctatttctagctctaaaac). 

The PCR reactions to amplify the sgRNA template were prepared using 3′ UTR sgRNA forward primer (FLA-1BP 3′SgRNAF: gaaattaatacgactcactataggGAGTGTGTGCTTTTCTCCCG**gttttagagctagaaatagc** or GP72 3′SgRNAF: gaaattaatacgactcactataggGTCATTTCCCACCTTTCTCC**gttttagagctagaaatagc**) and G00 universal SgRNA reverse primer as described previously [37,38]. For amplification of sgRNA templates, 0.2 mM dNTPs, 0.2 µM each of G00 and FLA-1BP 3′SgRNAF, and 0.02 U/µL Phusion™ High–Fidelity DNA Polymerase (Invitrogen, Carlsbad, CA, EUA) were mixed in 1× reaction buffer with MgCl_2_, 50 µL total volume. PCR conditions: 30 s at 98 °C, followed by 35 cycles of 10 s at 98 °C, 30 s at 60 °C, 15 s at 72 °C, and final stage for 5 min at 72 °C and 10 min at 4 °C. 

The donor DNA fragment containing the protein tagging 3xMyc::mNeonGreen::3xMyc (called mNG Tag) and the resistance gene was amplified by mixing of pPOTc-Blast-Blast-3xMyc::mNG::3xMyc [37] (DNA template containing the resistance gene of Blasticidin and the cMyc and mNeonGren tags) with either FLA-1BP or GP72 downstream forward and reverse primers (P4 FLA-1BPF: AAAGAAGAGCAACGTGCACGAGACATGTACggttctggtagtggttccgg/P5 FLA-1BPR: GAGTCAGAGAGACACAGAGACAGACAGAGAccaatttgagagacctgtgc and P4 GP72F: CGGAGCGCTGTGATTCTTGTTCCACCCATGggttctggtagtggttccgg/P5 GP72R: TGTTGGTTTCTTCTAATTTCAATTTCTGAAccaatttgagagacctgtgc, respectively). The PCR reactions were performed as reported previously [37]. Primer pairs 4 and 5 were used for C-terminal tagging. The primer P4 forward comprises a 30-nucleotide homology arm that hybridizes with the 3′UTR region, immediately preceding the SpCas9 PAM cleavage site, along with a region that binds to the pPOTc-Blast-Blast-3xMyc::mNG::3xMyc. The P5 reverse primer consists of a 30-nucleotide homology arm located in the 3′UTR region of the gene, downstream of the SpCas9 PAM site, along with a region that binds with the plasmid. For amplification of the donor DNA the following constituents were used and mixed in 1× reaction buffer with MgCl_2_ (100 µL total volume): 0.2 mM dNTPs, 0.2 µM each of P4 forward and P5 reverse, 200 ng pPOTc-Blast-Blast-3xMyc::mNG::3xMyc, and 0.02 U/µL Phusion™ High–Fidelity DNA Polymerase (Invitrogen, Carlsbad, CA, EUA). PCR conditions: 5 min at 98 °C followed by 45 cycles of 30 s at 98 °C, 30 s at 65 °C, 2 min and 15 s at 72 °C, and final stage for 5 min at 72 °C and 10 min at 4 °C. The donor DNA for tagging consists of a repair cassette with 30 nt homology flanks specific to the target locus, a 3xc-Myc::mNeonGreen::3xc-Myc tag, and blasticidin resistance, flanked by 5′ and 3′-untranslated region (UTR) sequences.

Constructs were transfected in epimastigotes of the T7Cas9 line as described previously [36,37,39]. Each cell line was analyzed using fluorescence microscopy, flow cytometry, and Western blot. The tags were called TcGP72::mNG and TcFLA-1BP::mNG.

### 2.4. Generation of TcGP72^−/−^ and TcFLA-1BP^−/−^ Double Knockout Cells via CRISPR/Cas9 Editing and Genotyping

To obtain knockout lines, we prepared two PCRs for amplification of SgRNA templates for each target gene as described in item 2.3. One sgRNA template was directed at 5′ UTR upstream of the target coding sequence (5′SgRNA FLA-1BP: gaaattaatacgactcactataggGTAAAGGAATTCTACACGTG**gttttagagctagaaatagc**; 5′SgRNAGP72: gaaattaatacgactcactataggGTAGACCGGACGTCCCGTCT**gttttagagctagaaatagc**), and one was directed 3′ UTR downstream (3′SgRNA FLA-1BP e 3′SgRNA GP72), as described previously for endogenous tagging [37]. Each double knockout was generated using two drug resistance markers (blasticidin and hygromycin), followed by cloning cell lines after transfection. 

To amplify the donor DNA fragment containing the resistance gene, we used DNA templates (pPOTc-Blast-Blast-Myc::mNG::myc or pPOTc-Hygro-Hygro-Myc::mNG::myc) mixed with a 5′ UTR forward primer upstream of the coding sequence and a 3′ UTR reverse primer downstream (P1 FLA-1BP forward: AAAGACGGGGGGGGGGGAGGTGCGGCAGGAgtataatgcagacctgctgc/P7 FLA-1BP reverse: GAGTCAGAGAGACACAGAGACAGACAGAGAccggaaccactaccagaacc and P1 GP72 forward: ACAAACAAACAAGCAAAAACAAAAATATTTgtataatgcagacctgctgc/P7 GP72 reverse: TGTTGGTTTCTTCTAATTTCAATTTCTGAAccggaaccactaccagaacc, respectively). Primer pairs 1 and 7 were used for knockouts. The P1 forward primer comprises a 30-nucleotide homology arm that hybridizes with the 5′UTR region, upstream of the cleavage site the SpCas9 PAM cleavage site, along with a region that binds to the pPOTc-Blast-Blast-3xMyc::mNG::3xMyc. The P7 reverse primer consists of a 30-nucleotide homology arm located in the 3′UTR region of the gene, downstream of the SpCas9 PAM site, along with a region that binds with the plasmid. The donor DNA for knockout consists of a repair cassette with 30 nt homology flanks specific to the target locus upstream and downstream and blasticidin resistance, flanked by 5′ and 3′-untranslated region (UTR) sequences. 

The PCR reactions for donor DNA amplification were performed as described [37,40] in item 2.3. Constructs were transfected in epimastigotes of the T7Cas9 line as described previously [36,37,39]. 

At 48 h after nucleofection, a selection marker was added to the cells (25 µg/mL blasticidin and 200 µg/mL hygromycin). Upon the establishment of selected cultures, the parasites were cloned through serial dilution. To confirm the gene deletions, genomic DNA (gDNA) was isolated using a Genomic DNA Mini Kit and used in PCR reactions with oligonucleotides that allowed the amplification of the 5′ end and part of the coding sequence of TcFLA-1BP (C1 FLA_1BP 5′UTR forward: TTCAGCAATAAGAAGAAGGAAGGG and C2 FLA1_BP ORF reverse: TACCTTCATAGAGCTTCACATCGG) and TcGP72 (C1 GP72 5′UTR forward: AAGAAGAGAGAGGGAGAGAGAG and C2 GP72 ORF REV: AAATGCCGGATTTGTTCACCAAAC) genes, as well as coding regions of the resistance genes for blasticidin (B1_BSD_Fow: CCTCATTGAAAGAGCAACGGC and B2_BDS_Rev AGGGCAGCAATTCACGAATC) and hygromycin (H1_Hygro_Fow: GAAAAAGCCTGAACTCACCGC and H2_Hygro_Rev: GTGTCGTCCATCACAGTTTGC). The PCR products were electrophoresed in 1% agarose gel. The result was detected using ethidium bromide staining, and visualization was performed with the UVP bioimaging system.

### 2.5. Western Blotting

Whole parasite extracts were prepared by collecting cells and boiling them in Laemmli sample buffer (60 mM Tris. HCl, pH 7.4, 1% *w*/*v* SDS, 10% *v*/*v* glycerol, 20 mM DTT, 0.05% *w*/*v* Bromophenol Blue). Extracts were resolved on 8% polyacrylamide gel and then transferred to nitrocellulose membranes (GE Healthcare Life Sciences, Piscataway, NJ, EUA) with standard procedures. The membranes were stained with 0.3% *w*/*v* Ponceau S in 0.3% *v*/*v* acetic acid, washed in water, and blocked with 5% *w*/*v* non-fat dry milk in Tris-Buffered Saline (10 mM Tris-HCl, pH 7.4, 0.15 M NaCl) with 0.05% Tween-20 (TBS-T). Primary antibody incubation was performed for 1 h at room temperature with TBS-T supplemented with 5% non-fat dry milk. 

Membranes were washed with TBS-T and then incubated with secondary antibodies in the same conditions as the primary antibodies. HRP-conjugated secondary antibodies (anti-rabbit or anti-mouse Promega, 1:300) were used for protein detection and visualization. The blot was revealed using ECL plus (Promega Corporation, 2800 Woods Hollow Road, Madison, WI 53711, EUA) and registered with an ImageQuant™ LAS 500 (GE Healthcare Life Sciences, Piscataway, NJ, EUA). Proteins were detected with Western blotting using the following antibodies: anti-Myc antibody 9E10 (Invitrogen, Carlsbad, CA, EUA, 1:3000 dilution), anti-aldolase antibody (1:3000) [41], anti- glycoprotein 82 (GP82) antibody (Mab 3F6, 1:1000) [35], and anti-eif5A [42].

### 2.6. Flow Cytometry of Fluorescent Parasites

Log-phase epimastigotes (1 × 10^6^ cells/mL) were washed and resuspended in phosphate-buffered saline (PBS 1X—137 mM NaCl, 2.7 mM KCl, 10 mM Na_2_HPO_4_, 2 mM KH_2_PO_4_, pH 7.4) and then examined with flow cytometry. The mNeonGreen (mNG) fluorescence was measured with flow cytometry using a BD Accuri™ C6 apparatus (BD Biosciences, La Jolla, CA, EUA). The mNG fluorescence was measured in the FL1-A channel (525/30 filter), and the data were analyzed with FlowJo v10 software. Negative controls including cells without mNG expression were used to draw gates to discriminate positive and negative events, and 20,000 events were analyzed per sample.

### 2.7. Localization of Protein Tags In Vivo and Immunofluorescence Microscopy

Cell cultures (1 × 10^7^ cells) of TcFLA-1BP::mNG and TcGP72::mNG carrying the 3xMyc::mNG::3xMyc tag and the wild type were washed in PBS 1X, incubated with Hoechst 33,342 dye (Life Science/Biotechnology, Carlsbad, CA, USA) in PBS for 10 min, washed in PBS 1X, and adhered to poly-lysine coverslips for 15 min. Live cells were then visualized in an Axio Observer Z1 (Carl Zeiss, Oberkochen, Baden-Württemberg, Germany) equipped with a Zeiss N-Achroplan 100×/1.25 Ph3 M27 objective and a CCD camera (AxioCam HR R3). TcGP72^−/−^ and TcFLA-1BP^−/−^ mutants and T7Cas9 were washed in PBS 1X, fixed with recently prepared 4% formaldehyde in PBS 1X for 10 min, and adhered to poly-lysine-coated coverslips for 15 min. Coverslips were washed with PBS 1X, the cells were permeabilized with PBS 1X + 0.1% Triton X-100 for 10 min, and then blocking was performed with 3% *w*/*v* BSA for 1 h at room temperature (RT). 

The parasites were incubated for 1 h at 4 °C with a 1:200 dilution of mAb L3B2 monoclonal antibody, which recognizes the FAZ-1 protein in the FAZ filament in *T. brucei* [43]. They were then washed three times with PBS 1X + 0.05% Tween-20 and incubated for 1 h at 4 °C with the secondary anti-mouse antibody conjugated with Alexa Fluor-488 (Thermo Fisher Scientific, Waltham, MA, USA) at 1:500. The coverslips were washed three times and incubated once more with blocked solution for 16 h at 4 °C. Next, parasites were washed three times with PBS 1X + 0.05% Tween-20 and incubated with monoclonal antibody mAb 2F6 (1:200), which recognizes a flagellar protein of ~70 kDa [44], and anti-mouse secondary antibody conjugated with Alexa Fluor-594 (Thermo Fisher Scientific, Waltham, MA, USA) at 1:500 under the same conditions described above. Coverslips were mounted with Prolong Mounting Medium with DAPI and visualized in Axio Observer Z1 (Carl Zeiss, Oberkochen, Baden-Württemberg, Germany) equipped with a Zeiss N-Achroplan 100×/1.25 Ph3 M27 objective and a CCD camera (AxioCam HR R3). 

### 2.8. Cell Cycle Analysis

Synchronization of epimastigote forms of *T. cruzi* in the G1/S phase of the cell cycle was achieved using hydroxyurea (HU). Cells in the exponential growth phase were incubated with 15 mM of HU for 16–24 h and then released by washing twice with PBS and suspending them in a culture medium with 10% FSB. Cell culture continued for 24 h, and samples were taken at 0, 6, 12, and 24 h post-treatment and processed. For flow cytometry analysis, 3 × 10^5^ cells were harvested with centrifugation, washed with PBS 1X, and fixed in 70% ethanol at −20 °C for 30 min. Then, they were washed once with PBS and suspended in a staining solution (69 μM propidium iodide, 38 mM citrate buffer pH 7.4, 0.2 mg.mL^−1^ RNase). 

The DNA content of propidium iodide-stained cells was analyzed with flow cytometry using a BD Accuri™ C6 apparatus (BD Biosciences, La Jolla, CA, EUA). Percentages of cells at different phases of the cell cycle were evaluated with Cyflogic software version 1.2.1. The control consisted of cells in the log growth phase without HU, and 20,000 events were analyzed per sample.

### 2.9. Ultrastructural Analysis

For scanning electron microscopy (SEM) analyses, cells were fixed for 1 h in 2.5% glutaraldehyde diluted in cacodylate buffer (0.1 M and pH 7.2). They were then adhered to coverslips, pre-coated with poly-L-lysine, and post-fixed for 1 h with 1% osmium tetroxide diluted in cacodylate buffer. Samples were dehydrated in a graded ethanol series (50%, 70%, 90%, and two exchanges of 100% ethanol for 10 min each step), and critical-point dried using CO_2_. Specimens were coated with a 5 nm layer of platinum and then visualized in an EVO 10 and ZEISS FIB-SEM AURIGA 40 scanning electron microscope at the multiuser unit of the National Center for Structural Biology and Bioimaging (CENABIO) at UFRJ. Cell lengths were measured in the SEM images using the AxioVision4 program. 

For transmission electron microscopy (TEM), parasites were washed twice in PBS and fixed in 2.5% *w*/*v* glutaraldehyde in 0.1 M cacodylate buffer at pH 7.2 for 1 h. Then, cells were washed in 0.1 M cacodylate buffer (pH 7.2) and post-fixed for 1 h using an osmium-thiocarbohydrazide-osmium (OTO) protocol [18,45]. After post-fixation, samples were washed in water, dehydrated in a series of increasing acetone concentrations, and embedded in epoxy resin. Ultrathin sections were obtained using an Ultracut Reichert Ultramicrotome and mounted on 400-mesh copper grids. Samples were stained with uranyl acetate and lead citrate and then analyzed using a Hitachi HT 7800 electron microscope operating at 100 kV at the CENABIO multiuser unit at UFRJ.

For negative staining, cells were harvested with centrifugation, adhered to formvar-coated grids, and extracted in 1% *v*/*v* NP-40 in PEME buffer (100 mM PIPES, 1 mM MgSO_4_, 0.1 mM EDTA and 2 mM EGTA, pH 6.9) for 5 min. They were then fixed in 2.5% (*v*/*v*) glutaraldehyde in PEME for 10 min as described previously [46]. Grids were negatively stained with 0.7% (*v*/*v*) aurothioglucose in water and observed in a Hitachi HT 7800 electron microscope operating at 100 kV at the CENABIO multiuser unit at UFRJ.

### 2.10. Growth Curve

To determine the growth rates of the cell lines, 5 × 10^6^ cells/mL (in the logarithmic phase) were seeded in 5 mL of LIT medium with 10% FBS and maintained at 28 °C for 7 days. The parasites were counted daily in a Neubauer chamber. 

### 2.11. Morphometric and Statistical Analyses

Morphometric analyses were conducted using SEM images and then processed using the ImageJ program V 1.8.0 (National Institutes of Health, Bethesda, MD, USA). Statistical analyses were carried out using GraphPad Prism software v6. *p*-values less than 0.05 were considered statistically significant. All figures were prepared in Affinity Designer 2 software Version 2.0.

### 2.12. Gene IDs

Orthologous and paralogous genes related to TcGP72 deposited in TritrypDB: *Angomonas deanei* strain Cavalho ATCC PRA-265 (ID: ADEAN_001018200—hypothetical protein, conserved), *Crithidia fasciculata* strain Cf-Cl (ID: CFAC1_040014300.1—flagellar glycoprotein-like protein), *Leishmania amazonensis* MHOM/BR/71973/M2269 (ID: LAMA_000162700.1—hypothetical protein, conserved), *Leishmania braziliensis* MHOM/BR/75/M2904 (ID: LbrM.10.0770—flagellum adhesion protein 1), *Leishmania donovani* CL-SL (ID: LdCL_100013600—flagellar glycoprotein-like protein), *Leishmania infantum* JPCM5 (ID: LINF_100013100—flagellar glycoprotein-like protein), *Leishmania major* strain Friedlin (ID: LmjF.10.0630—flagellum adhesion protein 1), *Trypanosoma brucei* EATRO1125 (IDs: Tb1125.8.4010—flagellum adhesion protein 1, Tb1125.8.4060—flagellum adhesion protein 2, and Tb1125.8.4110—flagellum adhesion protein 3), *Trypanosoma evansi* strain STIB 805 (ID: TevSTIB805.8.4130—flagellum adhesion glycoprotein), *Trypanosoma grayi* ANR4 (DQ04_03901040—flagellum adhesion glycoprotein), and *Trypanosoma vivax* Y486 (TvY486_0803430—flagellum adhesion protein 1).

Genes related to TcFLA-1BP: *Angomonas deanei* strain Cavalho ATCC PRA-265 (ID: ADEAN_001018300—hypothetical protein, conserved), *Crithidia fasciculata* strain Cf-Cl (ID: CFAC1_040014200.1—hypothetical protein, conserved), *Leishmania amazonensis* MHOM/BR/71973/M2269 (ID: LAMA_000162600.1—hypothetical protein, conserved), *Leishmania braziliensis* MHOM/BR/75/M2904 (ID: LbrM.10.0760—FLA1-binding protein), *Leishmania donovani* CL-SL (ID: LdCL_100013500—LdCL_100013500), *Leishmania infantum* JPCM5 (ID: LINF_100013000—hypothetical protein—conserved), *Leishmania major* strain Friedlin (ID: LmjF.10.0620—FLA1-binding protein), *Trypanosoma brucei* EATRO1125 (IDs: Tb1125.8.4100—FLA1-binding protein), *Trypanosoma evansi* strain STIB 805 (ID: TevSTIB805.5.5150—hypothetical protein, conserved), *Trypanosoma grayi* ANR4 (DQ04_03901050—hypothetical protein) and *Trypanosoma vivax* Y486 (TvY486_0803420—FLA1-binding protein).

## 3. Results

### 3.1. In Silico Analysis and Protein Localization of TcGP72 and TcFLA-1BP

The sequences of the glycoprotein 72 (GP72) or FLA-1 and FLA1-binding protein (FLA-1BP) in trypanosomatids were obtained from the genome of the Dm28c 2018 strain deposited in TritrypDB. We selected these proteins based on the previous studies on *T. cruzi* [29,47] and *T. brucei* [25,27,48]. In *T. cruzi*, both genes are unique copies within the haploid genome (TcGP72 ID: C4B63_21g104 and TcFLA-1BP ID: C4B63_21g106). 

The phylogenetic tree constructed based on the protein sequences of trypanosomatids (Figure 1A) revealed their clustering into Trypanosoma species with ~44 to 47% identity to TcGP72 and ~36–46 to TcFLA-1BP. After a comparison with Leishmania and Angomonas species, we observed higher sequence divergence (~23 to 28% for TcGP72 and ~25 to 28% for TcFLA-1BP). Our analysis showed that TcGP72 contains a unique transmembrane region at the N-terminus, followed by an NHL-repeat domain (Figure 1B). This is different from *T. brucei*, which exhibits a C-terminus transmembrane and 16-amino acid tail that is essential for flagellum attachment [27,32]. However, TcFLA1BP presents an organization like that of *T. brucei*, with a signal peptide at the N-terminus, an extracellular NHL-repeat domain, and a transmembrane domain followed by a C-terminus 45-amino acid tail (C45).

To address the function of TcGP72 and TcFLA-1BP, we made fluorescent cell lines designated as TcGP72::mNG and TcFLA-1BP::mNG (Figure 1B) by inserting an mNeonGreen and c-Myc reporter in the C-terminus of the endogenous gene using a CRISPR/Cas9 system [37]. The expression of the fusion proteins TcGP72::mNG and TcFLA-1BP::mNG were determined with flow cytometry and Western blotting using a Myc antibody (Figure 2). The flow cytometry experiments (Figure 2A) revealed differences in fluorescence intensity between tagged lines cells and the T7Cas9 control. TcFLA-1BP::mNG had a lower fluorescence intensity than TcGP72::mNG. In the blots of the protein extracts from epimastigotes (Figure 2B), we identified a single band, which migrated above the expected weight in both tagged proteins. The estimated weights for the tagged proteins were 98.2 kDa for TcGP72::mNG and 116 kDa for TcFLA-1BP::mNG.

The localization of the tagged proteins was monitored in live cells with intrinsic mNG-tag fluorescence and Hoechst staining. The TcGP72::mNG mutant displayed a reticulated labeling pattern (Figure 3A(c,d)), and the TcFLA-1BP::mNG mutant displayed labeling along the flagellum attachment region (Figure 3A(e,f)). To determine whether the localization of TcFLA-1BP is exclusively along the cell body of the parasite, we generated a TcFLA-1BP::mNG line with depletion of TcGP72 called TcFLA-1BP::mNGΔGP72 (Figure 3A(g,h)). As previously reported [15,29], upon TcGP72 depletion, flagellum detachment from the cell body occurred, and TcFLA-1BP::mNG was found only on the side of the cell body. The TcGP72::mNG parasite (Figure 3B) exhibited a morphological change similar to that of the TcGP72 knockout with a detached flagellum [15], which was not observed in the TcFLA-1BP::mNG mutant.

The mNeonGreen tag at the C-terminus led to protein retention in the endoplasmic reticulum, resulting in a dominant-negative phenotype like the knockout parasite for TcGP72. When assessing the localization of TcFLA-1BP::mNG in intracellular amastigotes and cell culture-derived TCTs with immunodetection (Figure 4), we observed anti-mNeonGreen labeling in the flagellar adhesion region, as marked by the monoclonal antibody L3B2, which specifically targets the FAZ-1 protein component of the FAZ filament of the flagellar adhesion zone (FAZ) in *T. brucei* [43]. This confirmed the labeling of the protein in the FAZ in all forms of the parasite.

### 3.2. Knockout of TcGP72 and TcFLA-1BP Affects T. cruzi Cell Morphology and Cell Cycle Progression of Epimastigotes

To investigate the impact of TcGP72 and TcFLA-1BP knockout throughout the life cycle of *T. cruzi*, we generated double knockout cells using the CRISPR/Cas9 system [37,39]. T7Cas9 epimastigotes were transfected with a mix of SgRNA templates corresponding to 5′SgRNA and 3′SgRNA, which mark the cleavage sites of Cas9 and donor DNAs containing resistance genes (blasticidin [BSD] or hygromycin [Hyg]), as well as the homology arms for each gene. After selection and cloning, we genotyped the parasites (Figure 5) to confirm gene editing. Both proteins were edited, and the amplification using primers annealed at the 5′UTR of the gene and inside the ORF (P1 and P2 for TcGP72 or TcFLA-1BP) showed amplification only in the control (734 bp TcGP72 and 654 pb TcFLA-1BP). 

The insertion of the resistance mark (blasticidin and hygromycin) at the locus was also confirmed using amplification with a set of primers that bind to the 5′UTR (P1 TcGP72 or TcFLA-1BP) of the endogenous gene and the ORF of the resistance gene (P5 BSD or HYG). We only observed amplification of both markers in the null mutants for TcGP72^−/−^ (856 pb BSD and 1656 pb Hyg) and TcFLA-1BP (953 pb BSD and 1089 pb Hyg).

Immunofluorescence microscopy using anti-FAZ-1 *T. brucei* (L3B2 antibody) and anti-flagellum (2F6 antibody) revealed morphological differences between parental and knockout lines (Figure 6A). In TcGP72^−/−^ epimastigotes, we observed flagellum detachment and shortening of the FAZ, which generated a parasite with a morphology resembling a promastigote (Figure 6B) and caused the formation of rosettes (Figure 7A). L3B2 labeling in this parasite was limited to the flagellar pocket, and 2F6 showed an intense signal in the same region, suggesting protein retention at this site. 

SEM and TEM analysis of TcGP72^−/−^ parasites showed protuberances and projections along the flagellum in this region (Figure 7A,B). Some TcGP72^−/−^ cells displayed twists filled with amorphous material in the flagellum (Figure 7A). TEM and negative staining analysis (Figure 6C) of the TcGP72^−/−^ line confirmed the flagellum detachment and the disappearance of the FAZ region along the parasite body. In the TcFLA-1BP^−/−^ line (Figure 6), there was a mixed population with fully detached or partially detached flagella. However, in most of them, the flagella were attached to their cell bodies, and it was possible to observe short FAZs (Figure 6B,C).

Based on the drastic changes in morphology, we made detailed measurements of cell body size and flagellum length for both null mutants (Figure 6D) using the SEM images. In TcGP72^−/−^ mutants, flagellar length exhibited dispersion with the appearance of cells possessing both longer and shorter flagella compared to control cells, resulting in variable lengths ranging from 3 to 16 μm. In the TcFLA-1BP^−/−^ mutant, we observed a reduction in flagellar length with a higher concentration of parasites having a flagellum smaller than 5 µm. Cell body size was considerably reduced in the TcGP72 null mutant (mean 7.491 µm), while the TcFLA-1BP^−/−^ (mean 9.352 µm) mutant remained similar to the control (9.592 µm). 

The ratio of flagellum length to cell size showed a pronounced increase in TcGP72^−/−^ and a slight decrease in TcFLA-1BP^−/−^ relative to the control. These results suggest that TcGP72 and TcFLA-1BP are involved in the coordination of flagellum/FAZ assembly in different ways. During cell growth, knockout mutant epimastigotes form rosettes. Considering this, we assessed the growth capacity of null mutants for 7 days through a growth curve under agitation at 400 rpm at 28 °C. Besides presenting drastic morphological changes, the null mutant epimastigotes for TcGP72^−/−^ and TcFLA-1BP^−/−^ grew slower compared to the T7Cas9 parasites.

### 3.3. Deletion of TcGP72 and TcFLA-1BP Alters Cell Cycle Progression

The cell cycle progression of TcGP72^−/−^ and TcFLA-1BP^−/−^ parasites was analyzed with flow cytometry of propidium iodide (PI)-labeled cells (Figure 8A). We observed a displacement of all main peaks in knockout parasites, which was most pronounced in the TcFLA-1BP^−/−^ line. The peak corresponding to the G1 phase of the cell cycle represents ~40% of the parasites in control cells (T7Cas9). At this stage, the DNA content corresponds to one nucleus (2n) and one kinetoplast. In knockout lines, there was a drastic decrease in G1 by 1.8% for TcGP72^−/−^ and 1% for TcFLA-1BP^−/−^. 

The S phase situated in the valley between the two peaks also showed a reduction in knockout parasites (~5% for TcGP72^−/−^ and 3.5% for TcFLA-1BP^−/−^) compared to the control cells (~12.8%). We observed a 2-fold increase in parasites in the G2/M phase (second peak) in null mutants (~93% TcGP72^−/−^ and 95% TcFLA-1BP^−/−^) compared to the control (~48%). In this phase, parasites have double the DNA content (4n) and undergo cytokinesis.

Given that the arrest in G2/M was significant, we synchronized cells at the G1/S transition using HU and analyzed them with flow cytometry. Both knockouts strains showed a delay in synchronization time compared to the parental line (Appendix A), returning to G1 only 6 h after treatment with HU (6 h pHU). At the 12 h point (12 h pHU) after HU release, we noticed a long peak in G1/S in the knockout lines compared to the control, and it was more pronounced in TcGP72^−/−^. At 24 h pHU, we observed that the majority of the knockout strain’s population was in G2/M, whereas approximately 50% of the control parasites were in G1/S.

Afterward, we evaluated the cell cycle progression by counting parasites stained with Hoechst with a microscope to estimate the percentage of parasites in each cell cycle (Figure 8B). We also examined the morphology of growing parasites using SEM (Figure 8C). Consistent with our previous results, we confirmed an increased percentage of parasites in cytokinesis, presenting atypical cellular patterns in knockout parasites. We observed aberrant, multi-flagellated parasites with non-segregating kinetoplasts and multinucleated cells. An increased number of diving cells was identified, and it was common to observe cells with adhered flagella, particularly in the TcGP72 knockout. 

### 3.4. Knockout of TcGP72 and TcFLA-1BP Affects T. cruzi Cell Morphology during Metacyclogenesis

To investigate the ability of the TcFLA-1BP and TcGP72 knockout to differentiate from epimastigotes to MTs in vitro, we incubated late log-phase epimastigotes in TAU3AAG medium for 96 h. MTs were collected from the culture supernatant following purification in a DEAE-cellulose column. The knockout parasites of TcGP72 and TcFLA-1BP exhibited a significantly reduced differentiation yield into MTs compared to the control parasites. The purification yield of TcGP72 was 38 times lower than that of the control; for TcFLA-1BP, it was 12 times lower. 

An intriguing morphological distinction was observed between T7Cas9 and null mutant MTs (Figure 9). Both knockout parasites exhibited flagellum detachment, resembling epimastigote forms. For TcGP72^−/−^, detachment was complete, while TcFLA-1BP^−/−^ displayed distinct FAZ patterns (Figure 9A). During metacyclogenesis, a substantial portion of TcGP72^−/−^ parasites remained attached to the culture substrate. We subjected these parasites to morphometric evaluations through SEM analyses (Figure 9B). We observed that the TcGP72^−/−^ parasites adhered to the bottom of the culture flask displayed a complex network of interactions between the parasites involving both the flagella and cell bodies, as well as between the parasites and the substrate. 

Ultrastructural analyses using TEM on purified MTs of TcGP72^−/−^ revealed a detached flagellum, an elongated nucleus, and a characteristic topological arrangement of the kDNA filamentous network typical of MTs (Appendix A). To confirm whether the purified parasites exhibited molecular characteristics of MTs, we conducted a Western blot analysis using antibodies against GP82, a stage-specific protein. Upon challenging the lysates of the purified parasites (Appendix A), we observed similar labeling in all the parasites. This finding indicates that despite their distinct morphology, both the TcGP72^−/−^ and TcFLA-1BP^−/−^ parasites exhibit classical molecular features of metacyclic forms.

## 4. Discussion

To better understand the biology of *T. cruzi* and gather insights for the prevention and treatment of Chagas disease, there is a growing focus on investigating its adaptability mechanisms. The number of investigations on *T. cruzi*’s adaptability mechanisms is increasing. This is vital to understand its biology and generate information for the prevention and treatment of Chagas disease. The various developmental stages of the parasite along its life cycle are designated based on the relative position of various structures, such as the nucleus, the kinetoplast, the flagellum’s emergence, and its association with the cell body. It is now clear that these morphological changes occur due to the modifications in the three-dimensional arrangement of the subpellicular microtubules [3,7,8,49]. 

The subpellicular microtubules maintain growth polarization towards the posterior end of the cell, and the other structures are rearranged according to the phase of the cycle [8,49,50]. *T. cruzi* has three basic stages (epimastigotes, TCTs, and amastigotes). In addition, intermediate stages have been recently identified [51]. The epimastigote forms are characterized by a long flagellum that emerges from the anterior portion and is attached to the cell body with a small free portion. It presents a rod-shaped kinetoplast, followed by a rounded nucleus. TCTs have a long flagellum emerging from the anterior region, which is also attached to the cell body with a free portion, followed by a rounded posterior kinetoplast and an elongated nucleus. Amastigote forms exhibit a shortened anterior flagellar structure, rod-shaped kinetoplast, and rounded nucleus [3,7,52].

In all forms of the parasite, the flagellum that emerges from the cell is attached to the cell body by either a long area (epimastigotes or TCTs) or a short area (amastigotes). The flagellum/cell body connection is established via the FAZ, an adhesive network composed of a set of structures that extends from the flagellar domain and interacts with the cytoskeleton. The FAZ is like a zipper with a portion originating from the skeleton and flagellar membrane, and enters the cell body membrane. It is fixed in the microtubule quartet (MtQ) antiparallel to the subpellicular microtubules. This structure is composed of a network of interconnected proteins, forming an electron-dense adherent network. The FAZ has been previously described as a “cell ruler” of morphology because it plays a crucial role in regulating cell size, organelle position, cell division, differentiation, and pathogenesis [23,24,26].

In this work, we characterized the structural and molecular organization of *T. cruzi*’s FAZ. When searching for the proteins identified in the FAZ of *T. brucei*, we observed that *T. cruzi* presents fewer FAZ proteins. Of the 34 proteins previously described in *T. brucei* [25], we identified only 24 orthologues in *T. cruzi*. Moreover, *T. brucei* presents isoforms not identified in *T. cruzi*. It is possible that the higher number of FAZ proteins in *T. brucei* may be evolutionarily advantageous for parasites evading the host immune system, since it lives in the bloodstream and the extracellular portion of some tissues. In *T. brucei*, it has been proposed that the FAZ plays a role in pathogenesis. The attachment of the flagellum to the parasite body allows host immune system factors deposited in this region to be directed to the flagellar pocket, where they can be ingested with endocytosis, degraded, and reabsorbed by the parasite [53].

We decided to focus on two proteins. First, the previously characterized TcGP72 protein presents some open gaps regarding its localization in the FAZ and internal ultrastructural changes during the cell cycle. Second, FLA-1BP is a possible ligand of GP72, which was already described in *T. brucei*. To characterize these proteins, we used the CRISPR/Cas9 system to generate mutants with endogenous genes labeled at the C-terminus with a fluorescent protein and knockout parasites for both proteins. By evaluating the sequences of the proteins in the different trypanosomatids, we observed conservation in the gene regions in which the proteins were inserted. The gene cluster comprises endonuclease G, FLA-1BP, FLA-1, and zinc finger protein. This clustering is observed in *T. cruzi*, *T. brucei*, *Crithidia fasciculata*, *Leishmania* spp., *T. gray*, *T. vivax*, and *T. evansi*, and is a great indication of the relevance of this region in the genome of various trypanosomatids.

TcGP72 and TcFLA-1BP exhibit a similarity of ~45% with the other trypanosome species, having a higher divergence (~25%) with *Leishmania* spp., *C. fasciculata*, and *Angomonas deanei*. Apparently, in *T. cruzi*’s different published genomes, it is impossible to find isoforms of TcGP72 as described in *T. brucei* (TbFLA-1, TbFLA-2, and TbFLA-3) [27,31]. Interestingly, the isoforms in the *T. brucei* genome also exhibit the same clustering as FLA-1. By analyzing the domains present in each protein, we noticed that in *T. cruzi*, TcGP72 presents a different organization from that observed in all other trypanosomatids. 

In *T. brucei*, both TcFLA-1 and TcFLA-1BP proteins show similar organization with the short intracellular domain (N-terminus)—a signal peptide, a long extracellular domain containing NHL repeats, and transmembrane followed by an intracellular segment (C-terminus), with 16 amino acids (C16) for TbFLA-1 and 40 amino acids (C40) for TbFLA-1BP [27,32]. We observed that TcGP72 has a different arrangement and does not contain the second transmembrane domain at the C-terminus. Among all trypanosomatids evaluated, *T. cruzi* is the only one that presents this arrangement. TcFLA-1BP has the same arrangement as *T. brucei* but with a slightly longer C-terminus segment with 45 amino acids. In the model proposed for *T. brucei*, the C-terminus segment is important for anchoring the FAZ in the flagellum and cell body [27]. 

In the mutants with proteins truncated at the C-terminus, we observed that TcGP72 was retained in the cytoplasm and probably in the endoplasmic reticulum. TcFLA-1BP showed a tagging in the cell membrane different from that proposed in *T. brucei* in the flagellar membrane [27]. This localization in the cell body was confirmed when we generated in the TcFLA-1BP-tagged mutant, the TcGP72 knockout, which showed detachment from the flagellum, and TcFLA-1BP::mNG remained bound to the parasite body. Our results are the first to show that this model is reversed in *T. cruzi*. The localization of TcGP72 in the flagellar domain of the FAZ was also reported previously using small labels, such as HA, Ty, and Flag [15,29,47,54]

In the model that we propose for *T. cruzi* (Figure 10), TcGP72 crosses the flagellar membrane, with the short intracellular domain facing the flagellum lumen, where it is probably attached. The extracellular domain faces the intermembrane region. On the side of the cell body, TcFLA-1BP is anchored by the intracellular domains at the N- and C-terminus, and the extracellular domain faces the intercellular domain of the FAZ. This allows the proteins to potentially interact with the NHL domains.

The increase in molecular weight bands of the labeled TcGP72::mNG and TcFLA-1BP::mNG proteins can be explained by the addition of glycoconjugates to proteins. It has previously been demonstrated in *T. cruzi* that TcGP72 is a highly glycosylated protein. Our findings support the previously published data for TcGP72 and indicate that TcFLA-1BP can also undergo post-translational modifications [55,56]. In our in silico analyses conducted with the bioinformatics tools NetOGlyc 4.0 and NetNGlyc 1.0. https://services.healthtech.dtu.dk/ (accessed on 24 February 2023), we identified 21 possible sites of O-glycosylation and 2 sites of N-glycosylation for TcGP72, as well as 3 O-glycosylation sites and 4 N-glycosylation for TcFLA-1BP.

An important finding in this work was the morphological alteration that was discovered to be similar to a knockout in the TcGP72::mNG parasite. The TcGP72::mNG protein retained in the cytoplasmic structures generated a dominant-negative result in an alteration in the protein transfer to the flagellar membrane and lead to a detachment of the flagellum as well as the TcGP72 knockout. These data demonstrate the necessity of correct folding and the importance of the C-terminus for transferring the protein to the flagellar membrane. The expression of TcFLA-1BP::mNG was observed throughout the FAZ in all cell cycle forms of *T. cruzi* and was lower in the amastigotes due to the short FAZ.

The deletion of TcGP72 caused flagellum detachment and the disappearance of the FAZ connections, which is consistent with the previous reports on *T. cruzi* [15,29,54]. Morphologically, the parasite showed a shortening of the cell body, with a variety of flagellar sizes, including parasites with long flagella (12 and 15 µM) and short flagella (1 and 4 µM) compared to the control. However, in the TcGP72 knockout parasites, the flagellum/cell body ratio appeared scattered, resulting in a culture with a homogeneous morphology resembling promastigote forms of Leishmania. 

In the TcFLA-1BP knockout parasite, we observed a heterogeneous population with fully or partially detached flagella. Most parasites showed short flagella, while their cell body sizes remained similar to the control parasite. In parasite knockouts for TcFLA-1BP, we commonly found parasites with a reduced FAZ exhibiting lateral attachment to the parasite body. In *T. brucei*, RNAi of FLA-1BP causes complete flagellum detachment and a reduction in the length of the FAZ and cell body, resulting in epimastigote-like parasites [27].

Labeling with L3B2 (anti-FAZ-1) revealed an accumulation of labeling near the flagellar pocket for all forms in the TcGP72 knockout. In epimastigotes and TCTs, this finding was more evident, and the same accumulation was observed when tagging for flagellar proteins as Mab2F6 antibodies. The knockout for TcFLA-1BP showed a milder accumulation of these proteins, probably due to the partial detachment of the FAZ structure.

The FAZ and flagellum assembly dynamics are polarized and require initial nucleation, where proteins directed to the region near the flagellar pocket are anchored to the membrane and are concomitantly extended with flagellum assembly [57,58]. In the model proposed for *T. brucei*, the FAZ proteins are carried in vesicles that accumulate at the proximal end of the FAZ near the flagellar pocket, where they are inserted into the membrane of the flagellum and cell body. As subunits are added, they cause displacement of the other inserted proteins, leading to an extension of the structure along the flagellum [25,58]. 

We observed that the nucleation of proteins to the proximal region of the flagellar pocket continues even in the absence of the FAZ in both knockouts. However, targeting is mainly affected with the TcGP72 knockout, which presents protein accumulation in the proximal region, large protrusions, and flagellar projections. We observed that even with a detached flagellum, the TcFLA-1BP::mNGΔGP72 parasite was able to direct the TcFLA-1BP::mNG protein to the FAZ region in the cell body. Our findings suggest that even without the FAZ connections, the flagellum is assembled; however, its extent is altered, probably due to impaired intraflagellar transport (IFT) along the axoneme.

Cells depleted of TcGP72 and TcFLA-1BP showed failure in proper cell cycle progression. In epimastigotes, knockout parasites showed growth retardation with generation time that was approximately two times smaller than the control’s time. When evaluating DNA replication and cytokinesis, we observed drastic alterations, with knockout parasites presenting a variable number of nuclei, kinetoplasts, and flagella compared to the controls. This asymmetric division of the knockout parasites impaired the standard cell cycle progression because, when evaluating the parasites with flow cytometry, we identified that the absence of both proteins results in parasite accumulation in the G2/M phase of the cell cycle, with characteristics of parasites undergoing cytokinesis. 

Cell cycle delay was observed even after HU synchronization. The altered division was also observed with light microscopy and SEM. In *T. brucei*, the deletion of most of the FAZ component proteins causes altered cytokinesis, and long culture times cause cell death. Most studies on *T. brucei* are performed in an inducible editing system [28,31,43]. In our analyses, we observed that TcGP72 and TcFLA-1BP knockouts remained viable, even with a reduced growth rate and impaired cytokinesis, as previously described [29,55].

In trypanosomatids, it has been described that the changes in the three-dimensional arrangement of the cytoskeleton enable the repositioning of intracellular structures of the parasites [8,25,49]. The dynamics of formation and rearrangement of subpellicular and cytoplasmic microtubules allow for the correct cell division of the cell body and other organelles and are thus directly linked to the morphological changes between forms [49]. The FAZ in trypanosomes creates a seam between the quartet of specialized microtubules with the surrounding subpellicular microtubules, thus directing cell growth [8,48,50]. 

The deletion of the TcGP72 and TcFLA-1BP proteins resulted in morphological changes and alterations in the positioning of cytoplasmic structures (mainly the nucleus and kinetoplast). During metacyclogenesis, we could not identify the differentiation rates because we lost the reference of the anterior and posterior regions of the parasites. However, we noticed that in the purified MTs, there were parasites with elongated shapes, globular kinetoplasts, and elongated nuclei that showed a reaction against the GP82 antigen characteristics of MTs [35]. 

We hypothesized that even without repositioning organelles, the parasite could remodel the cellular structures and membrane components characteristic of each form while maintaining differentiation. Somehow, subpellicular and cytoplasmic microtubules rearrange themselves to maintain an architecture that allows structural remodeling of various structures and organelles, such as the cytostome–cytopharinx complex, FAZ, nucleus, and kinetoplast.

## Figures and Tables

**Figure 1 pathogens-12-01367-f001:**
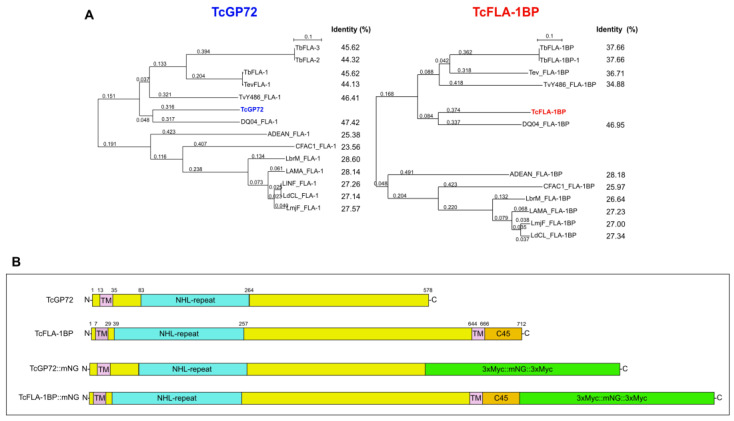
Bioinformatics analysis and domain organization of TcGP72 and TcFLA-1BP. (**A**) Phylogenetic tree from protein sequences of trypanosomatids. Amino acid sequences were aligned using Clustal Omega https://www.ebi.ac.uk/Tools/msa/clustalo/ (accessed on 22 April 2020, and the aligned sequences were used to generate a phylogenetic tree (SeaView 5.4). (**B**) Diagram of the conserved domains found in TcGP72 and TcFLA-1BP orthologous to *Trypanosoma brucei*. TcGP72 contains a transmembrane domain (TM) at the N-terminus and an NHL-repeat in the extracellular domain. TcFLA-1BP is structured with a transmembrane domain (TM) at the N-terminus, followed by NHL-repeat-containing extracellular domain, TM, and a 45-amino acid tail (C45) at the C-terminus. Numbers indicate the amino acid residues. The domains in the diagram were labeled based on the information deposited in the TritrypDB database. Domain organization is shown for TcGP72 and FLA-1BP fused to c-Myc and mNeonGreen tags at the C-terminus (3xMyc::mNG::3xMyc), named TcGP72::mNG and TcFLA-1BP::mNG. Tb: *Trypanosoma brucei*. Tc: *Trypanosoma cruzi*. Tev: *Trypanosoma evansi*. TvY486: *Trypanosoma vivax*. DQO04: *Trypanosoma grayi*. ADEAN: *Angomonas deanei*. CFAC1: *Crithidia fasciculata*. LbrM: *Leishmania braziliensis*. LAMA: *Leishmania amazonensis*. LINF: *Leishmania infantum*. LdCL: *Leishmania donovani*. LmjF: *Leishmania major strain*.

**Figure 2 pathogens-12-01367-f002:**
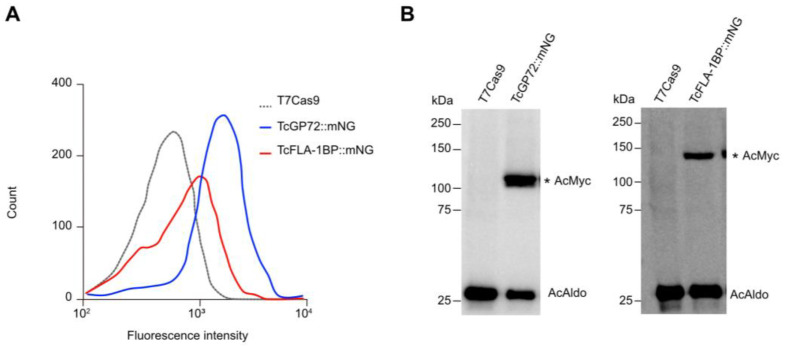
Analysis of the expression of fluorescent proteins. (**A**) Representative histograms showing the expression of fluorescent proteins in the mutant parasites. The peak with a black dashed line represents T7Cas9 lineage (control), and peaks with blue and red lines show TcGP72::mNG and TcFLA-1BP::mNG parasites, respectively. (**B**) Western blot analysis of fluorescent parasite (epimastigote) lysates expressing C-Myc and mNeonGreen fusion proteins (equivalent to 10^7^ cells). Blots were probed with anti-Myc (AcMyc) antibodies and re-probed with anti-aldolase (AcAldo). kDa: molecular weight marker. The asterisk (*) shows the tagged protein bands. TcGP72::mNG: 98.2 kDa. TcFLA-1BP::mNG 116 kDa. TcAldolase: 29.3 kDa.

**Figure 3 pathogens-12-01367-f003:**
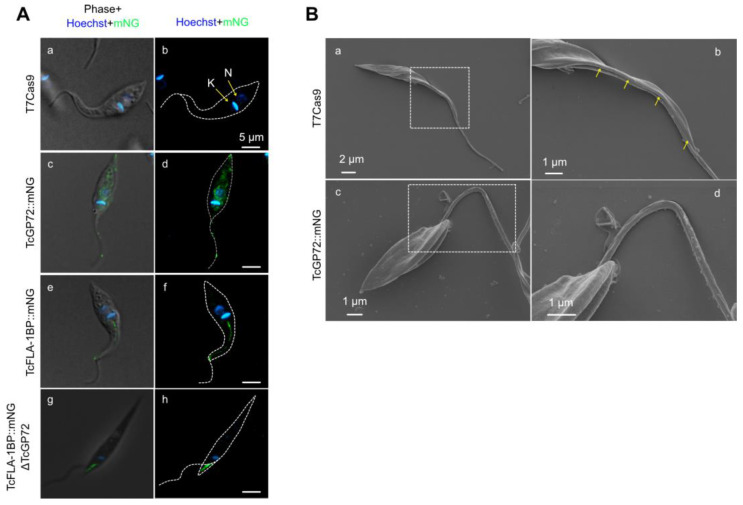
Localization of TcGP72::mNG and TcFLA-1BP::mNG proteins. (**A**) The fluorescent parasites generated with CRISPR/Cas9 were analyzed with fluorescence microscopy. Direct mNeonGreen fluorescence (green) reveals the cellular distribution of TcGP72::mNG (**c**,**d**) and TcFLA-1BP::mNG (**e**,**f**). TcGP72::mNG localized with distributed reticulated on the cell body and flagellum. In contrast, TcFLA-1BP::mNG was localized in just the FAZ. To confirm the FAZ domain localization of the TcFLA-1BP, we performed TcGP72 knockout in the TcFLA-1BP::mNG parasite (TcFLA-1BP::mNGΔGP72). TcFLA-1BP::mNGΔGP72 (**g**,**h**) parasites showed the detached flagellum and tagging exclusively on the side of the cell body along the FAZ. DAPI for DNA (blue). K: kinetoplast, N: nucleus. (**B**) Scanning electron microscopy shows TcGP72::mNG parasite (**c**,**d**) morphology with flagellum detachment in contrast to the T7Cas9 control parasite (**a**,**b**). The dotted region shows the magnified area, and the yellow arrows show the FAZ.

**Figure 4 pathogens-12-01367-f004:**
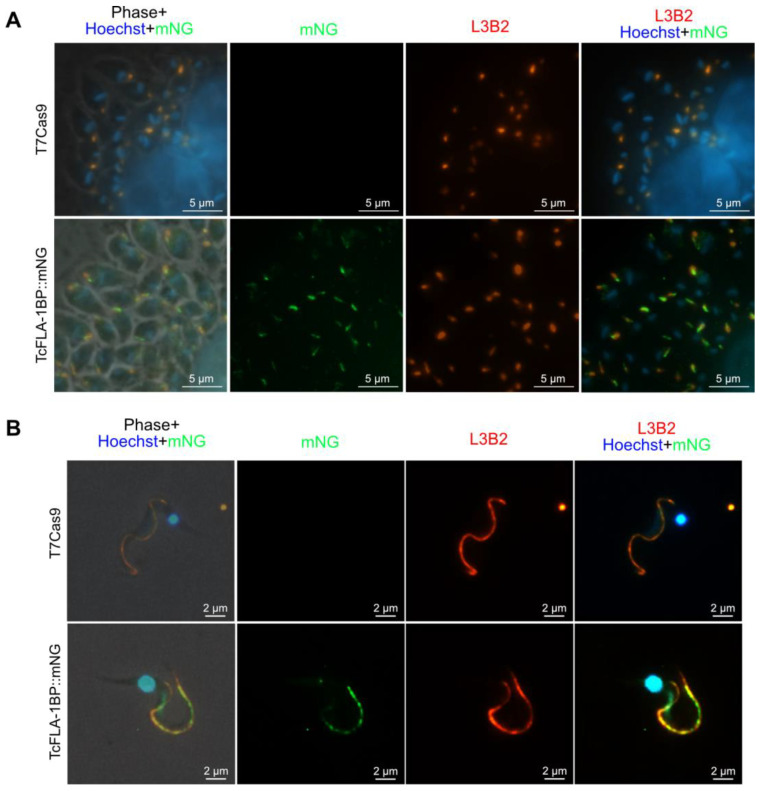
TcFLA-1BP targets the FAZ in amastigotes and trypomastigote (TCT) forms. (**A**) Fluorescence microscopy of TcFLA-1BP::mNG and T7Cas9 control in intracellular amastigotes (IAs) with FAZ labeled with anti-L3B2 (FAZ-1 *T. brucei*; red) and anti-mNG (mNeonGreen tag; green). (**B**) Localization in tissue culture derived TCTs. Nuclei and kinetoplasts were stained with Hoechst (blue). Scale bars: 2 and 5 µm.

**Figure 5 pathogens-12-01367-f005:**
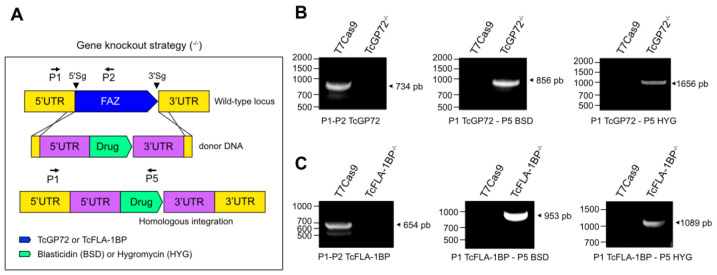
Depletion of TcGP72 and TcFLA−1BP. Illustration of gene knockout strategy (not to scale). (**A**) Diagram showing the TcGP72 and TcFLA-1BP locus and PCR primers (arrows) used to confirm the presence of the TcGP72 and TcFLA-1BP coding sequence (blue box) or the correct integration of the drug-resistant genes (green box). An arrowhead marks Cas9 cleavage sites. Yellow and purple boxes represent the untranslated regions (UTRs) of endogenous genes and donor DNAs. (**B**,**C**) PCR products were visualized on agarose gel. T7Cas9, parental cell line; TcGP72^−/−^, double knockout parasite to TcGP72; TcGP72^−/−^, double knockout parasite to TcFLA-1BP. P1 (forward primer) anneals to the 5′ UTR of TcGP72 or TcFLA-1BP. P2 (reverse primer) anneals to the coding sequence of TcGP72 or TcFLA-1BP. P5 (reverse primer) anneals to the end of the resistance gene coding sequence (blasticidin or hygromicin).

**Figure 6 pathogens-12-01367-f006:**
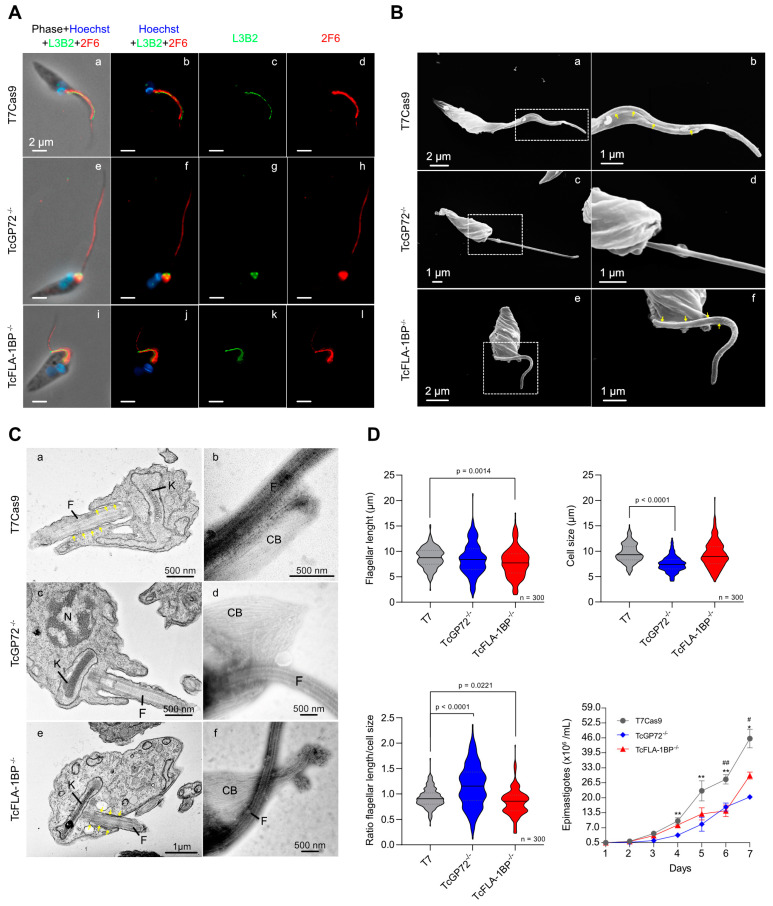
TcFLA-1BP and TcGP72 knockout in *T. cruzi* causes partial or total flagellum detachment and impairs assembly of the flagellum, morphogenesis, organelle positioning, and cell growth. (**A**) Fluorescence microscopy of TcGP72::mNG and TcFLA-1BP::mNG with FAZ labeled with anti-L3B2 (FAZ-1 *T. brucei*; green) or anti-2F6 (PFR; red). Nuclei and kinetoplasts were stained with Hoechst (blue). Scale bars: 2 µm. T7Cas9 (**a**–**d**). TcGP72^−/−^ (**e**–**h**). TcFLA-1BP^−/−^ (**i**–**l**). (**B**) SEM showing the morphology of knockout parasites. TcGP72^−/−^ (**c**,**d**) with total flagellum detachment and TcFLA-1BP^−/−^ (**e**,**f**) partial detachment relative to the T7Cas9 control parasite (**a**,**b**). (**C**) TEM (**a**,**c**,**e**) and negative staining TEM (**b**,**d**,**f**) analysis. (**a**,**c**,**e**) Axial section of epimastigote forms showing the cell body (CB), kinetoplast (K), nuclei (N), flagellum (F), and FAZ (yellow arrows). (**b**,**d**,**f**) Negatively stained epimastigote forms showing the subpellicular microtubule of the cell body (CB) with a fully detached flagellum as in TcGP72^−/−^ or a partially detached flagellum as in TcFLA-1BP^−/−^. (**D**) Violin plots show flagellum length, cell size, and the relation between flagellum length and cell size in the epimastigote forms. The mean values are indicated with a black line. Statistically significant differences are indicated with *p*-values. n = 300 cells. The line chart depicts the growth curves of the knockout parasites. Statistically significant differences are indicated with asterisks for TcGP72^−/−^ and pound signs for TcFLA-1BP^−/−^ in relation to the T7Cas9 control (* or # *p* < 0.05 and ** or ## *p* < 0.001).

**Figure 7 pathogens-12-01367-f007:**
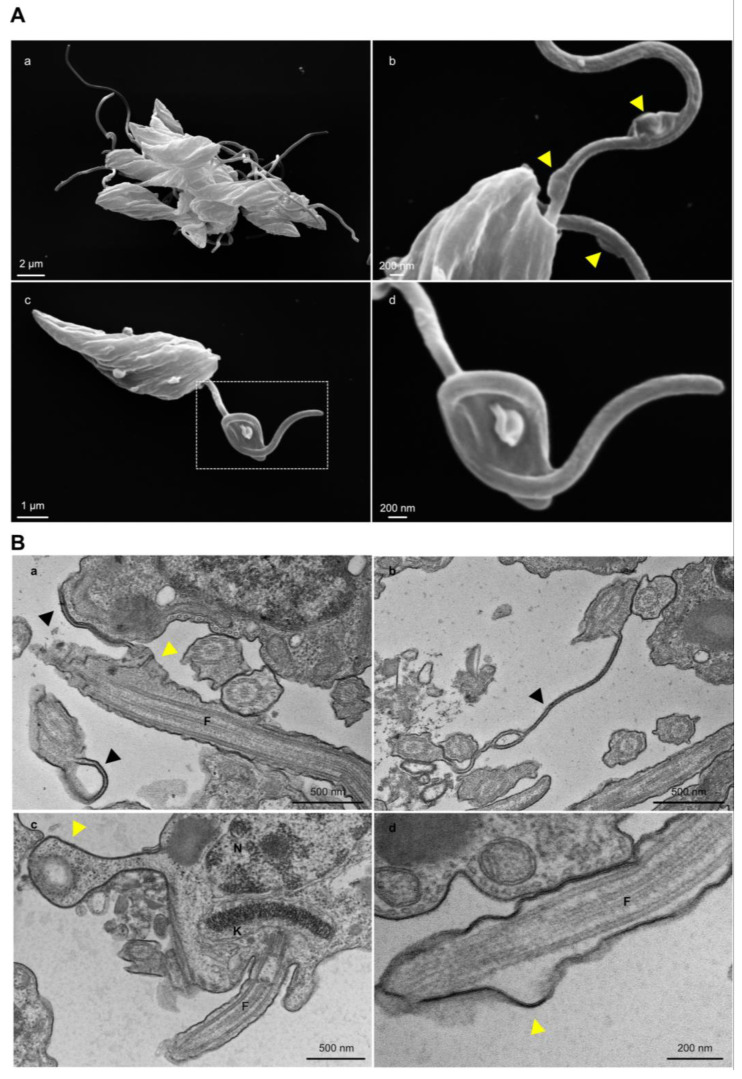
TcGP72 knockout parasites display rosette formation, protrusions, and projection along the flagellum. (**A**) SEM of TcGP72^−/−^ epimastigotes. (**a**) Rosette formation. (**b**) Flagellar membrane protrusions (yellow arrowhead) and (**c**,**d**) flagellar structures bizarrely filled with amorphous material. The dotted region (**c**) shows the magnified area (**d**). (**B**) TEM shows (**a**,**b**) the membrane projections (black arrowhead) and (**a**,**c**,**d**) protrusions (yellow arrowhead) of the flagellum and the cell body membrane in TcGP72 knockout parasites. F: Flagellum.

**Figure 8 pathogens-12-01367-f008:**
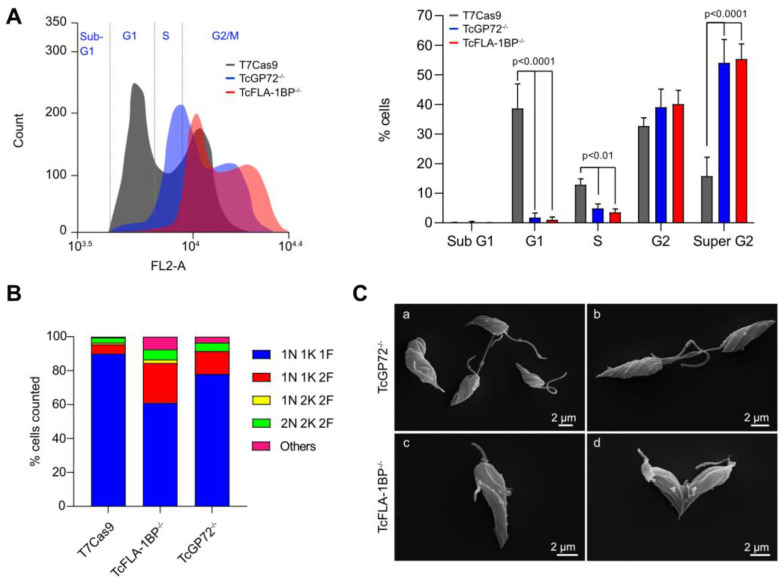
TcGP72 and TcFLA-1BP knockout in *T. cruzi* epimastigotes show cell cycle defects. Cell cycle comparative analysis with flow cytometry of the knockout parasites. (**A**) Representative histograms showing the DNA content of TcGP72^−/−^, TcFLA-1BP^−/−^, and the T7Cas9 control. The peak in gray represents the T7Cas9 parasites and peaks in red and blue represent depleted TcFLA-1BP and TcGP72, respectively. The bar graph represents the percentage of cells in each cell cycle phase in three independent experiments. Statistical analysis was performed with two-way ANOVA with Bonferroni correction for multiple testing. (**B**) Quantification of nuclei, the kinetoplast, and the flagellum in cells labeled for immunofluorescence microscopy with anti-2F6 (PFR) and Hoechst stain of knockout parasites (n = 300 cells). (**C**) SEM showing the abnormal morphology of knockout parasites. TcGP72^−/−^ (**a**,**b**) and TcFLA-1BP^−/−^ (**c**,**d**).

**Figure 9 pathogens-12-01367-f009:**
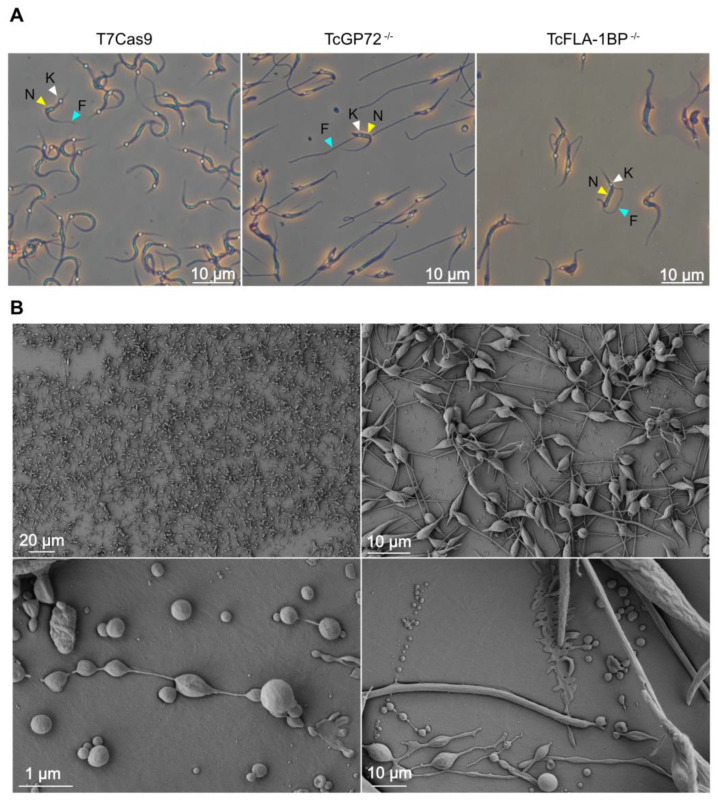
TcGP72 and TcFLA−1BP knockout *T. cruzi* show metacyclogenesis defects. (**A**) Metacyclic TCTs were purified on a DEAE cellulose column stained with Giemsa and observed under a microscope. The white arrowhead represents the kinetoplast, the yellow arrowhead represents the nucleus, and the blue arrowhead represents the flagellum. (**B**) Scanning microscopy images of TcGP72^−/−^ parasites forming an adhesion network at the bottom of the culture flask during the metacyclogenesis assay.

**Figure 10 pathogens-12-01367-f010:**
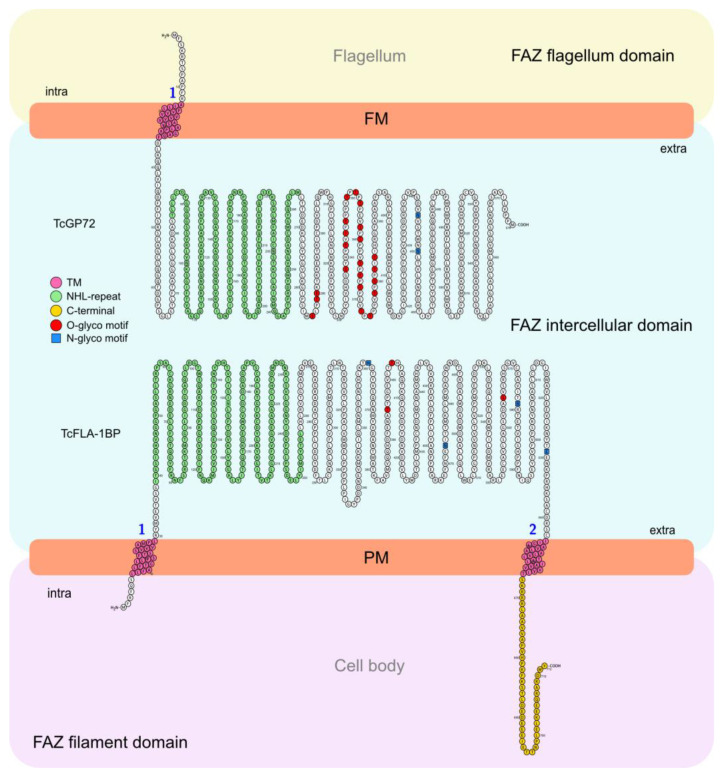
A simplified view of the flagellum and cell body adhesion mediated by the interaction of TcGP72 and FLA-1BP in *Trypanosoma cruzi*. In *T. cruzi*, the flagellum membrane interacts with the cell body membrane, forming an adhesive network of proteins. This region is called the flagellar attachment zone (FAZ). The FAZ is divided into 3 main domains: (1) FAZ flagellar domain, (2) FAZ intercellular domain, and (3) FAZ filament domain. The components of the FAZ are distributed along the domains. Both TcGP72 and FLA1BP are essential for this adhesion at different intensities. In our model, TcGP72 is present on the flagellar membrane, and TcFLA-1BP is present on the cell membrane. TcGP72 is anchored to the flagellum by the intracellular domain at the N-terminus. TcFLA-1BP is anchored to the cell body by the two intracellular domains (N- and C-terminus). TcGP72 and TcFLA-1BP interact via the extracellular domains directed towards the intercellular domain of FAZ, thereby mediating flagellum/cell body adhesion. The numerals 1 and 2, highlighted in blue, denote the transmembrane regions. This interaction happens via the NHL-repeat domains present in the extracellular regions. The absence of TcGP72 and TcFLA-1BP causes partial or complete detachment of the flagellum. Both proteins have N- and O-glycosylation sites. Extra: extracellular. Intra: intracellular. FM: flagellum membrane. PM: plasma membrane.

## Data Availability

All data related to this work are presented in this manuscript.

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
