# Peer review of "Implications of Flagellar Attachment Zone Proteins TcGP72 and TcFLA-1BP in Morphology, Proliferation, and Intracellular Dynamics in Trypanosoma cruzi"

_pathogens, 2023, doi:10.3390/pathogens12111367_

Round 1

Reviewer 1 Report

Comments and Suggestions for Authors

this an excellent work, with very good images that deserve o be published

Two items to be reviewed are: First a better description of the molecular biology experiments, perhaps as supplementary material.

Second. The References were not carefully edited and throughout the text, the taxonomic designation of T. cruzi  was not always italicised

Author Response

Response to Reviewer 1 X Comments

1. Summary

2. Questions for General Evaluation

Reviewer’s Evaluation

Response and Revisions

Does the introduction provide sufficient background and include all relevant references?

Yes/Can be improved/Must be improved/Not applicable

Are all the cited references relevant to the research?

Yes/Can be improved/Must be improved/Not applicable

Is the research design appropriate?

Yes/Can be improved/Must be improved/Not applicable

Are the methods adequately described?

Yes/Can be improved/Must be improved/Not applicable

Are the results clearly presented?

Yes/Can be improved/Must be improved/Not applicable

Are the conclusions supported by the results?

Yes/Can be improved/Must be improved/Not applicable

3. Point-by-point response to Comments and Suggestions for Authors

Comments 1: First a better description of the molecular biology experiments, perhaps as supplementary material.

Response 1: Thank you for your comments. We have reviewed the suggestions and incorporated the recommended changes into the molecular biology experiments, as outlined in the text at the specified locations:

·       Page 4 and lines 151 – 153: The design of the primers used for the amplification of the sgRNA template and donor DNA was conducted through the utilization of the LeishEdit platform (http://www.leishgedit.net/Home.html), using the genome of the T. cruzi strain DM28c.

·       Page 4 and line 159: (taatacgactcactatagg) for in vitro transcription of RNA.

·       Page 4 and lines 162-164: sgRNA primer designer: T7 RNAP promoter/PROTOSPACER/ region complementary RNA scaffold SpCas9. G00 universal SgRNA (aaaagcaccgactcggtgccactttttcaagttga-taacggactagccttattttaacttgctatttctagctctaaaac).

·       Page 4 and lines 169-173: For amplification of sgRNA templates, 0.2 mM dNTPs, 0.2 µM each of G00 and FLA-1BP 3’SgRNAF, and 0.02 U/µL Phusion™ High–Fidelity DNA Polymerase (Thermo Scientific) were mixed in 1x reaction buffer with MgCl2, 50 µL total volume. PCR conditions 30 s at 98°C followed by 35 cycles of 10 s at 98°C, 30 s at 60°C, 15 s at 72°C and final stage for 5 min at 72°C and 10 min at 4°C.

·       Page 4 and lines 176-177: (DNA template containing the resistance gene of Blasticidin and the cMyc and mNeonGren tags).

·       Page 4 and lines 183-196: Primer pairs 4 and 5 were used for C-terminal tagging. The primer P4 forward com-prises a 30-nucleotide homology arm that hybridizes with the 3’UTR region, immediately preceding the SpCas9 PAM cleavage site, along with a region that binds to the pPOTc-Blast-Blast-3xMyc::mNG::3xMyc and primer P5 reverse consists of a 30-nucleotide homology arm located in the 3'UTR region of the gene, downstream of the SpCas9 PAM site, along with a region that binds with the plasmid. For amplification of donor DNA were used, 0.2 mM dNTPs, 0.2 µM each of P4 forward and P5 reverse, 200 ng pPOTc-Blast-Blast-3xMyc::mNG::3xMyc of and 0.02 U/µL Phusion™ High–Fidelity DNA Polymerase (Thermo Scientific) were mixed in 1x reaction buffer with MgCl2, 100 µL total volume. PCR conditions: 5 min at 98°C followed by 45 cycles of 30 s at 98°C, 30 s at 65°C, 2 min and 15 s at 72°C and final stage for 5 min at 72°C and 10 min at 4°C. The donor DNA for tagging consists of a repair cassette with 30 nt homology flanks specific to the target locus, a 3xc-Myc::mNeonGreen::3xc-Myc tag, and blasticidin resistance, flanked by 5' and 3'-untranslated region ( UTR) sequences.

·       Page 5 and line 203: as described in item 2.3.

·       Page 5 and lines 218-226: Primer pairs 1 and 7 were used for knockouts. The primer P1 forward comprises a 30-nucleotide homology arm that hybridizes with the 5'UTR region, upstream of the cleavage site the SpCas9 PAM cleavage site, along with a region that binds to the pPOTc-Blast-Blast-3xMyc::mNG::3xMyc and primer P7 reverse consists of a 30-nucleotide homology arm located in the 3'UTR region of the gene, downstream of the SpCas9 PAM site, along with a region that binds with the plasmid. The donor DNA for knockout consists of a repair cassette with 30 nt homology flanks specific to the target locus upstream and downstream and blasticidin resistance, flanked by 5' and 3'-untranslated region (UTR) sequences.

Comments 2: The References were not carefully edited and throughout the text, the taxonomic designation of T. cruzi was not always italicized.

Response 2: We acknowledge the feedback provided. While implementing the revisions, we inadvertently overlooked certain formatting errors within the references and taxonomic nomenclature. Consequently, we have rectified these issues across the entire manuscript, incorporating additional references where necessary and ensuring the appropriate italicization of scientific names. Modifications to the citation format were implemented in the following pages.

·       Page 5 and line 208.

·       Page 6 and line 290.

·       Page 7 and line 332.

·       Page 8 and line 394.

·       Page 19 and line 694.

Reviewer 2 Report

Comments and Suggestions for Authors

The manuscript is well designed and brings interesting results on the characterization of Trypanosoma cruzi flagellar attachment zone (FAZ). Using the CRISPR/Cas9 system, the study characterizes FAZ-specific proteins and their unique organization in T. cruzi. Deletion of these proteins resulted in various effects on intracellular structures, cytokinesis, and metacyclogenesis, underscoring the significant impact of FAZ-related proteins on the parasite's division and differentiation.

Minor points

Increase the resolution of all figures. The low resolution made it difficult to see morphological alterations and fluorescence profiles.

Figure 2b, it is important to see the entire membrane showing the single band. The migration above the expected molecular weight could demonstrate a glycosylation. Is it possible to establish in silico or experimentally?

In Figure 2, legend, it is 107 cells or 107?

Figure 4 also needs improvements in quality, and fluorescent images are at low resolution and often pixelized. 

Figure 4a shows the colocalization plots; the fluorescence of merge images doesn't seem to colocalize.

Figure 6 (D) is a violin plot, not a dot plot. 

The flagellum is important for parasite attachment to the peri microvillar membrane in the vector. Possibly, the detachment of the flagellum could decrease the development of the parasite in the vector. In the metacyclogenesis assay, quantification of parasite form during differentiation was performed. This result could add to the manuscript discussion.

The authors think that FAZ proteins could be a target for chemotherapy.

Comments on the Quality of English Language

The English is reasonably well written, only the correction of some small typos is necessary.

Author Response

Response to Reviewer 2 X Comments

1. Summary

2. Questions for General Evaluation

Reviewer’s Evaluation

Response and Revisions

Does the introduction provide sufficient background and include all relevant references?

Yes/Can be improved/Must be improved/Not applicable

Are all the cited references relevant to the research?

Yes/Can be improved/Must be improved/Not applicable

Is the research design appropriate?

Yes/Can be improved/Must be improved/Not applicable

Are the methods adequately described?

Yes/Can be improved/Must be improved/Not applicable

Are the results clearly presented?

Yes/Can be improved/Must be improved/Not applicable

Are the conclusions supported by the results?

Yes/Can be improved/Must be improved/Not applicable

3. Point-by-point response to Comments and Suggestions for Authors

Comments 1: Increase the resolution of all figures. The low resolution made it difficult to see morphological alterations and fluorescence profiles.

Response 1: We appreciate the provided recommendations. We have conducted a thorough assessment of the image quality and believe they are now at their optimal quality. Should any concerns persist regarding image quality, please do not hesitate to communicate them. It is important to note that the images have been prepared at a resolution of 300 dpi by the magazine's requirements.

Comments 2: Figure 2b, it is important to see the entire membrane showing the single band. The migration above the expected molecular weight could demonstrate a glycosylation. Is it possible to establish in silico or experimentally? In Figure 2, legend, it is 107 cells or 107?

Response 2: Thank you for the suggestions. We have incorporated the suggested modifications: Figure 2b has been revised in accordance with the recommendation, as indicated in line 542, and the caption has been altered (page 12 line 147 - “107”). In silico analyses to assess glycosylation sites were conducted using bioinformatics tools. We have also revised the text section describing glycosylation sites for greater clarity. (page 20 line 147 - “107”)

·       Page 20 lines 727-729: In our in silico analyses conducted with the bioinformatics tools NetOGlyc 4.0 and NetNGlyc 1.0.(https://services.healthtech.dtu.dk/), we identified 21 possible sites of O-glycosylation and 2 sites of N-glycosylation for TcGP72 and 3 O-glycosylation sites and 4 N-glycosylation for TcFLA-1BP.

Comments 3: Figure 4 also needs improvements in quality, and fluorescent images are at low resolution and often pixelized. Figure 4a shows the colocalization plots; the fluorescence of merge images doesn't seem to colocalize.

Response 3: We agree with the suggestion to increase the resolution of Figure 4 (page 13 line 563) and acknowledge the observation of non-co-localization of the markers. The L3B2 antibody labels FAZ-1, a component of the flagellar adhesion zone (FAZ) filament in Trypanosoma brucei. Consequently, it serves as a reference for the FAZ in T. cruzi, and we cannot assert co-localization. We have made changes in the manuscript where we previously suggested co-localization.

·       Page 9 lines 416-419: “…we observed anti-mNeonGreen labeling in the flagellar adhesion region, as marked by the monoclonal antibody L3B2, which specifically targets the FAZ-1 protein component of the FAZ filament of the flagellar adhesion zone (FAZ) in T. brucei [43].”

Comments 4: Figure 6 (D) is a violin plot, not a dot plot.

Response 4: Thank you for the feedback, we have modified the caption of Figure 6. (page 13 line 563)

·       Page 15 lines 591: “…Violin plots show flagellum length…”

Comments 5: The flagellum is important for parasite attachment to the peri microvillar membrane in the vector. Possibly, the detachment of the flagellum could decrease the development of the parasite in the vector. In the metacyclogenesis assay, quantification of parasite form during differentiation was performed. This result could add to the manuscript discussion. The authors think that FAZ proteins could be a target for chemotherapy. The authors think that FAZ proteins could be a target for chemotherapy.

Response 5: The metacyclogenesis rates of the knockout parasites are very low compared to the control, as we discussed in lines 506-507. We had great difficulty recovering the metacyclic forms after differentiation in the TAU3AAG medium and purification in DEAE cellulose. As for quantification during differentiation, we encountered many obstacles. Trypanosoma cruzi stages are characterized by topological and morphological patterns of organelles and stage-specific molecular markers. With the detachment of the flagellum, we lost the reference of what is anterior and posterior of the parasite, as discussed in lines 709-805. We were able to identify the purified form as metacyclic-like only by labeling it with the stage-specific antibody (Ac-GP82). We faced numerous challenges in characterizing intermediate differentiation stages during metacyclogenesis due to our difficult in determining the organelle positions.

        We have initiated the in vitro characterization and are in the process of planning in vivo experiments involving triatomines and mice. Based on the data acquired thus far, we cannot definitively determine whether this protein constitutes a viable chemotherapeutic target, as the parasite appears to exhibit continued progression in its absence. We anticipate that forthcoming data will contribute to a more comprehensive discussion on this matter. Nevertheless, the FAZ proteins represent a promising candidate for a vaccine target, given their presence throughout all stages of the T. cruzi cell cycle.

4. Response to Comments on the Quality of English Language

Point 1: The English is reasonably well written, only the correction of some small typos is necessary.

Response 1: Thank you for the suggestion. We have implemented the necessary revisions.
